# High Fecal Carriage of Extended-Spectrum β-Lactamase Producing *Enterobacteriaceae* by Children Admitted to the Pediatric University Hospital Complex in Bangui, Central African Republic

Hugues Sanke-Waïgana [1,*], Cheikh Fall [2], Jean-Chrysostome Gody [3], Eliot Kosh Komba [3], Gilles Ngaya [1], Jean-Robert Mbecko [1], Brice Martial Yambiyo [4], Alexandre Manirakiza [4], Guy Vernet [1], Alioune Dieye [5] and Yakhya Dieye [2,6]

1. Laboratoire de Recherche en Bactériologie Médicale et Expérimentale, Institut Pasteur de Bangui, Bangui BP 923, Central African Republic
2. Pôle de Microbiologie, Institut Pasteur de Dakar, Dakar BP 220, Senegal
3. Complexe Hospitalier-Universitaire Pédiatrique, Bangui BP 911, Central African Republic
4. Unité d'Épidémiologie, Institut Pasteur de Bangui, Bangui BP 923, Central African Republic
5. Faculté de Médecine, de Pharmacie et d'Odonto-Stomatologie, Université Cheikh Anta Diop, Dakar BP 5005, Senegal
6. École Supérieure Polytechnique, Université Cheikh Anta Diop, Dakar BP 5085, Senegal
* Correspondence: hughes.sanke@pasteur-bangui.cf; Tel.: +236-72-50-44-42

**Abstract:** Antimicrobial resistance (AMR) is a global public health threat. Quality data on AMR are needed to tackle the rise of multidrug-resistant clones. These data are rare in low-income countries, especially in sub-Saharan Africa. In this study, we investigated the rise of extended-spectrum β-lactamase–producing (ESBL) *Enterobacteriaceae* in Bangui, Central African Republic. We collected 278 fecal samples from 0–5-year-old children admitted to the Pediatric University Hospital Complex in Bangui from July to September 2021. *Enterobacteriaceae* were isolated and identified, and their susceptibility to 19 antibiotics was tested. We recovered one and two *Enterobacteriaceae* species from 208 and 29 samples, respectively. One clone of each species from each sample was further characterized, for a total of 266 isolates. *Escherichia coli* predominated, followed by *Klebsiella*. AMR was frequent, with 98.5% (262/266) of the isolates resistant to at least one antibiotic. Additionally, 89.5% (238/266) of the isolates were multidrug resistant, with resistance being frequent against all tested antibiotics except carbapenems and tigecycline, for which no resistance was found. Importantly, 71.2% (198/278) of the children carried at least one ESBL species, and 85.3% (227/266) of the isolates displayed this phenotype. This study confirms the rise of ESBL *Enterobacteriaceae* in Bangui and stresses the need for action to preserve the efficacy of antibiotics, as crucial for the treatment of bacterial infections.

**Keywords:** fecal carriage; *Enterobacteriaceae*; ESBL; MDR; Bangui

## 1. Introduction

Antimicrobial resistance (AMR) has become a global public health threat. Multidrug-resistant (MDR) clones of important human pathogens are emerging at an alarming rate [1], and experts agree that if proper actions are not taken, the number of deaths due to resistant bacteria will increase in the future. A memorable wakeup call for the threat posed by AMR came in 2014, when Lord Jim O'Neill and his team estimated that AMR could cause 10 million deaths a year by 2050 [2]. Following this prediction, the United Nations (UN) General Assembly brought AMR into focus when it adopted a resolution affirming a strong political commitment and advocating a global, holistic, "One Health" approach to the problem. Several initiatives followed this action, designing strategies to efficiently address

AMR worldwide [3]. The production of quality data on AMR and on antimicrobial use through multi-sectorial surveillance systems and capacity building were key strategies. It was also obvious that low- and middle-income countries (LMIC) needed special attention in this global effort [4]. Indeed, high-income countries (HIC) have already established strong surveillance and monitoring systems that enable early detection and rapid response to outbreaks caused by multidrug-resistant bacteria [5]. In contrast, LMIC, especially in sub-Saharan Africa (sSA), still suffer knowledge gaps in the prevalence and spread of many life-threatening multidrug-resistant (MDR) pathogens [6]. Thus, quality data on AMR are lacking on many sSA countries [7].

β-lactams are the most widely used antibiotics in human and veterinary medicine [8,9]. They comprise four classes, including penicillins, cephalosporins, monobactams and carbapenems, and are efficient treatments against many bacterial pathogens including both Gram-positive and Gram-negative bacteria. Resistance to β-lactams is primarily mediated by β-lactamases (BLs), which are enzymes that inactivate β-lactams [8]. These enzymes constitute a large and diverse family of molecules for which two classifications are proposed [10]. The Ambler classification, based on amino acid sequence, groups BLs into four classes, including A, C and D, which are enzymes harboring a serine residue in their active site, while class B (metallo-β-lactamases) comprises enzymes that require one or two zinc ions for substrate hydrolysis [11]. The Bush-J classification is based on substrate and inhibitor profiles and divides BLs in three major groups that are relevant to the phenotypes of clinical isolates [10]. Extended-spectrum β-lactamases (ESBLs) are defined as BLs effective against third and fourth generation cephalosporins and monobactams, and are inhibited by β-lactamase inhibitors such as clavulanic acid, sulbactam and tazobactam [12]. The first known ESBLs were class A BLs belonging to the TEM and SHV families, which emerged in the 1970s from point mutations in the genes coding for the original enzymes and having a narrow spectrum [13]. Interestingly, these enzymes appeared a few years after the introduction of expanded-spectrum cephalosporins, to which they confer resistance, for the treatment of bacterial infections [12]. This underlines the impact of antimicrobial use in the emergence and dissemination of AMR. In the 1980s, CTX-M type ESBL emerged, a new subgroup of class A BLs, which rapidly disseminated worldwide and today constitutes the most prevalent ESBLs [14]. While class A enzymes are the most prevalent, other ESBLs belong to classes C and D in the Ambler classification [15]. Interestingly, derivatives of TEM- and SHV-type ESBLs that were resistant to the ancient inhibitors clavulanic acid, sulbactam and tazobactam emerged, challenging the initial definition of this BL family [15]. Today ESBLs, especially class A enzymes, have widely disseminated across the globe, likely because they are mostly encoded by genes located on plasmids or other mobile genetic elements [15].

Extended-spectrum β-lactamase–producing *Enterobacteriaceae* (ESBL-Eb) are an important public health threat in the global context of increased AMR [16]. Several *Enterobacteriaceae* species, such as *Salmonella* and *Shigella,* are important gastrointestinal pathogens. Others, including *Escherichia coli* and *Klebsiella*, are commensals of the human and animal gastrointestinal tract. However, these species also contain clones that are pathogenic, causing different diseases. The gastrointestinal tract is a rich ecological niche, where bacteria exchange genetic materials, including AMR determinants. ESBL genes, which are often located on plasmids, are particularly affected by this phenomenon, which favors their dissemination [17]. The importance of ESBL-Eb is exemplified by the choice of ESBL *E. coli* as the target for the Tricycle program, a worldwide surveillance project initiated by the World Health Organization (WHO) [18], and by the classification of *Enterobacteriaceae* resistant to third-generation cephalosporins as priority 1 pathogens for which urgent actions are needed [19]. In LMIC, poor hygiene conditions and improper use of antimicrobials are important risk factors for the dissemination of AMR and the selection of resistant clones, including ESBL producers [20]. In this study, we aimed to contribute quality data on AMR and address the rise of ESBL-Eb prevalence in the Central African Republic (CAR). We

report on the fecal carriage of ESBL-Eb by 0–5-year-old children admitted to the Pediatric University Hospital, Bangui, CAR.

## 2. Results

### 2.1. Demographic and Clinical Data

The cohort included 278 children composed of 125 (45%) and 153 (55%) males and females, respectively. These patients were suffering different illnesses, including malaria ($n$ = 155, 55.8%), respiratory ($n$ = 60, 21.6%) and gastric ($n$ = 28, 10.1%) diseases, anemia ($n$ = 28, 10.1%), and other infections ($n$ = 7, 2.5%) (Table 1). Classification of patients in five one-year interval age groups showed that the number of children 0–12 months old was higher in the whole cohort as well as in each disease group (Table 1). Bacterial carriage was independent of age (Pearson correlation coefficient = 0.017; $p$ = 0.783), gender (Pearson correlation coefficient = 0.070; $p$ = 0.244) and disease symptoms (Pearson correlation coefficient = −0.069; $p$ = 0.251).

**Table 1.** Demographic and clinical data of participants admitted to the children's hospital in Bangui.

| Diagnostic | Age Group (Months) | | | | | |
|---|---|---|---|---|---|---|
| | 0–12 | 13–24 | 25–36 | 37–48 | 49–60 | Total |
| Malaria | 69 | 21 | 17 | 17 | 31 | 155 |
| Respiratory diseases | 40 | 4 | 1 | 2 | 13 | 60 |
| Gastric diseases | 20 | 2 | 1 | 2 | 3 | 28 |
| Anemia | 10 | 5 | 7 | 2 | 4 | 28 |
| Other diseases | 6 | 1 | 0 | 0 | 0 | 7 |
| Total | 145 | 33 | 26 | 23 | 51 | 278 |

### 2.2. Escherichia coli and Klebsiella pneumonia Predominated in Fecal Carriage in Hospitalized Children from Bangui

Out of the 278 fecal samples analyzed, 237 (85.3%) were positive for one (87.8%, $n$ = 208) or two (12.2%, $n$ = 29) *Enterobacteriaceae* species (Table 2). *E. coli* was predominant, being present in 74.3% ($n$ = 176) of the positive samples, followed by *Klebsiella pneumonia* (26.6%, $n$ = 63), *Klebsiella ornithinolytica* (6.8%, $n$ = 16), and *Klebsiella oxytoca* (2.9%, $n$ = 7) (Table 2). *Citrobacter koseri* was found in one single and one co-carriage, while *Enterobacter aerogenes* and *Enterobacter sakazakii* were found each in one sample as a mono-carriage. Notably, *E. coli* was present in all co-carriages, with the second most frequent species in co-carriage being *K. pneumoniae*, isolated from 19/29 (65.5%) of these samples (Table 2).

**Table 2.** *Enterobacteriaceae* isolated from stools of participants at the children's hospital in Bangui.

| Diagnostic | Ec | Kp | Kor | Kox | Ck | Ea | Es | Ec + Kp | Ec + Kor | Ec + Kox | Ec + Ck | None | Total |
|---|---|---|---|---|---|---|---|---|---|---|---|---|---|
| Malaria | 83 | 24 | 5 | 3 | 1 | 0 | 1 | 5 | 5 | 1 | 1 | 26 | 155 |
| Resp. | 22 | 16 | 0 | 1 | 0 | 0 | 0 | 9 | 2 | 1 | 0 | 9 | 60 |
| Gastric | 20 | 1 | 3 | 0 | 0 | 0 | 0 | 3 | 0 | 0 | 0 | 1 | 28 |
| Anemia | 17 | 2 | 1 | 1 | 0 | 0 | 0 | 2 | 0 | 0 | 0 | 5 | 28 |
| Others | 5 | 1 | 0 | 0 | 0 | 1 | 0 | 0 | 0 | 0 | 0 | 0 | 7 |
| Total | 147 | 44 | 9 | 5 | 1 | 1 | 1 | 19 | 7 | 2 | 1 | 41 | 278 |

Resp., respiratory diseases; *Ec*, *Escherichia coli*; *Kp*, *Klebsiella pneumoniae*; *Kor*, *Klebsiella ornithinolytica*; *Kox*, *Klebsiella oxytoca*; *Ck*, *Citrobacter koseri*; *Ea*, *Enterobacter aerogenes*; *Es*, *Enterobacter sakazakii*.

### 2.3. Fecal Carriage of Multidrug-Resistant and ESBL-Producing Enterobacteriaceae in Hospitalized Children

All isolates were tested for susceptibility to 19 antibiotics belonging to seven classes (Supplementary Table S1). ESBL-producing isolates were recovered from 198 out of the 278 fecal samples analyzed, corresponding to 71.2% (198/278) of the children carrying ESBL-Eb (Table 3). Detailed analysis of this carriage showed that 60.8% (*n* = 169) and 10.4% (*n* = 29) of the children harbored at least one and at least two ESBL-producing species, respectively (Table 3). Analysis of the recovered isolates showed frequent resistance, with 262/266 (98.5%) clones resistant to at least one drug and 238/266 (89.5%) to MDR strains (Table 4). Resistance was high for most of the antibiotics tested, especially those belonging to the penicillin, cephalosporin and quinolone classes (Supplementary Table S1). Additionally, ESBL production as shown by growth on CHROMagar-ESBL plate and resistance to third- and fourth-generation cephalosporins revealed 227/266 (85.3%) isolates that displayed this phenotype (Table 4). Notably, all isolates were sensitive to imipenem, ertapenem, amikacin and tigecycline (Supplementary Table S1).

**Table 3.** Extended-spectrum β-lactamase–producing *Enterobacteriaceae* in fecal samples from children at the Pediatric Hospital Complex, Bangui.

|  | **No-ESBL** | **1-ESBL** | **2-ESBL** | **Total** |
|---|---|---|---|---|
| Mono-carriage | 39 | 169 | NA | 208 |
| Bi-carriage | 0 | 0 | 29 | 29 |
| Total | 39 | 169 | 29 | 237 |

Mono-carriage, samples where one species was isolated; Bi-carriage, samples where two species were isolated; No-ESBL, samples with non-ESBL-producing isolates; 1-ESBL, samples with one ESBL-producing isolate; 2-ESBL, samples with two ESBL-producing isolates.

**Table 4.** Multidrug-resistant and extended spectrum beta-lactamase–producing *Enterobacteriaceae* from participants at the children's hospital in Bangui.

| Phenotype | Number and Percentage of Isolates Displaying the Shown Phenotype | | | | | | | |
|---|---|---|---|---|---|---|---|---|
|  | **All** | **Ec** | **Kp** | **Kor** | **Kox** | **Ck** | **Ea** | **Es** |
| MDR | 238/266 (89.5%) | 152/176 (86.4%) | 61/63 (96.8%) | 16/16 (100%) | 6/7 (85.7%) | $\frac{1}{2}$ (50%) | 1/1 (100%) | 1/1 (100%) |
| ESBL | 227/266 (85.3%) | 143/176 (81.3%) | 60/63 (95.2%) | 15/16 (93.8%) | 5/7 (71.4%) | 2/2 (100%) | 1/1 (100%) | 1/1 (100%) |

All, all bacterial species; *Ec*, *Escherichia coli*; *Kp*, *Klebsiella pneumoniae*; *Kor*, *Klebsiella ornithinolytica*; *Kox*, *Klebsiella oxytoca*; *Ck*, *Citrobacter koseri*; *Ea*, *Enterobacter aerogenes*; *Es*, *Enterobacter sakazakii*.

Analysis of antibiotic resistance within species showed that *Klebsiella* were the most resistant, with 61/63 (96.8%), 16/16 (100%) and 6/7 (85.7%) MDR *K. pneumoniae*, *K. ornithinolytica* and *K. oxytoca*, respectively, and 60/63 (95.2%), 15/16 (93.8%) and 5/7 (71.4%) clones of the same species producing ESBLs (Table 4). In comparison, *E. coli*, the most frequently isolated species, included 152/176 (86.4%) and 143/176 (81.3%) MDR and ESBL-producing isolates, respectively (Table 4).

Analysis of resistance with respect to ESBL phenotype showed that ESBL-producing isolates were significantly more frequently resistant than non-ESBL-producing clones (Table 5). As expected, all ESBL isolates were resistant to third- and fourth-generation cephalosporins, including cefotaxime, ceftriazone and cefepime, as well as to the first-generation cephalexin (Table 5). Similarly, resistance to ceftazidime (third generation) was high, with only one susceptible clone, and 218 and 8 isolates with a resistant or intermediate phenotype, respectively. In contrast, all the non-ESBL strains were susceptible to the six cephalosporins tested (Table 5). Not surprisingly, all the analyzed isolates were resistant to ampicillin, and all the ESBL and most of the non-ESBL clones were resistant to ticarcillin, confirming the frequent resistance of *E. coli* to these antibiotics [16]. Interestingly, 47.1% of

the ESBL isolates were resistant to clavulanic acid (Table 6). This was observed in *E. coli*, *K. pneumonia*, *K. ornithinolytica*, *Enterobacter aerogenes* and *Enterobacter sakazakii* (Table 6), which suggests the presence of a gene encoding an inhibitor-resistant BL, which might be an ESBL in the corresponding isolates. Similarly to β-lactams, resistance to the other classes of antibiotics (aminoglycosides, quinolones and chloramphenicol) was significantly higher in ESBL compared to non-ESBL isolates (Table 5). This is expected, since ESBL genes are often located on plasmids, which additionally harbor other antibiotic resistance genes [12].

**Table 5.** Antibiotic resistance of *Enterobacteriaceae* isolates according to extended spectrum β-lactamase production phenotype.

| Antibiotics | ESBL * | Non-ESBL * |
|---|---|---|
| Ampicillin | 227 (100%) | 39 (100%) |
| Ampicillin + Clavulanic acid | 107/227 (47.1%) | 0 |
| Ticarcillin | 227 (100%) | 34/39 (87.2%) |
| Cephalexin | 227 (100%) | 0 |
| Cefoxitin | 53/227 (23.3%) | 0 |
| Cefotaxime | 227/227 (100%) | 0 |
| Ceftazidime | 226/227 (99.6%) | 0 |
| Ceftriazone | 227 (100%) | 0 |
| Cefepime | 227 (100%) | 0 |
| Netilmicin | 106/227 (46.7%) | 1/39 (2.6%) |
| Tobramycin | 169/227 (74.4%) | 2/39 (5.1%) |
| Gentamicin | 160/227 (70.5%) | 2/39 (5.1%) |
| Nalidixic acid | 209/227 (92.1%) | 28/39 (71.8%) |
| Ciprofloxacin | 206/227 (90.7%) | 12/39 (30.8%) |
| Chloramphenicol | 163/227 (71.8%) | 20/39 (51.3%) |

ESBL, extended-spectrum β-lactamase–producing isolates. The numbers correspond to isolates with resistant (R) or intermediate (I) phenotypes. *, Pearson Chi-Square analysis showed that the differences between ESBL and non-ESBL isolates were all statistically significant.

**Table 6.** Clavulanic acid resistance in extended-spectrum β-lactamase–producing and negative isolates.

| Phenotype | *Ec* | *Kp* | *Kor* | *Kox* | *Ck* | *Ea* | *Es* |
|---|---|---|---|---|---|---|---|
| EBSL + | 67/143 (46.9%) | 27/60 (45%) | 10/15 (66.7%) | 1/5 (20%) | 0/2 | 1/1 | 1/1 |
| ESBL − | 0/33 | 0/3 | 0/1 | 0/2 | None | None | None |

*Ec*, *Escherichia coli*; *Kp*, *Klebsiella pneumoniae*; *Kor*, *Klebsiella ornithinolytica*; *Kox*, *Klebsiella oxytoca*; *Ck*, *Citrobacter koseri*; *Ea*, *Enterobacter aerogenes*; *Es*, *Enterobacter sakazakii*. The numbers and percentages refer to clavulanic acid-resistant isolates. None, no ESBL-negative isolate recovered for this species.

## 3. Discussion

AMR is a global public health concern. Sub-Saharan Africa is the most affected region of the world [1], with high prevalence of resistant bacteria and a lack of laboratories properly equipped and staff skilled to perform and interpret antimicrobial susceptibility testing. Consequently, despite a wealth of reports on AMR, there is a scarcity of quality data that can support the design of evidence-based guidelines for the treatment of bacterial infections. In this study, we analyzed the resistance profile of *Enterobacteriaceae* isolated from fecal samples of children admitted to the children's hospital (Complexe Hospital-Universitaire Pédiatrique), Bangui, CAR. We included all patients regardless of disease diagnosis. Clearly, our approach did not aim to investigate the bacterial etiology of the diseases, but rather to analyze the fecal carriage of antibiotic-resistant Enterobacteriaceae. This approach has several advantages. Firstly, it gives insight into the fecal carriage of the isolated bacteria in the communities the patients come from. Indeed, most of the studies focusing on AMR in sSA report results of investigations of patients affected by bacterial infections, while data from the community are rare [21]. Secondly, studying fecal carriage is interesting in that

*Enterobacteriaceae* are predominant in feces and are associated with different bacterial infections [22], including gastrointestinal, urinary tract and invasive infections, for which knowledge of the antimicrobial susceptibility of the responsible species are of public health importance. Thirdly, *Enterobacteriaceae* are powerful vectors of AMR spread, since the gastrointestinal tract houses a luxurious flora that horizontally exchanges genetic elements, including vectors of antimicrobial-resistance determinants [23].

Not surprisingly, *E. coli* predominated in the isolates we recovered, followed by *K. pneumoniae*, with these two species constituting 66% of the co-carriages (Table 2). *E. coli* and *K. pneumoniae* are commensals of the human gut, and each species also include pathogenic clones responsible for different diseases. They have been described as the most frequent bacterial pathogens in studies conducted in both HIC and LMIC [24–26]. We were particularly interested in the fecal carriage of ESBL-Eb. Indeed, these bacteria are of major concern worldwide. They are associated with prolonged hospital stay and increased case-fatality rate. We found a high proportion of ESBL-producing strains, with 85.3% of the isolates displaying this phenotype. This is one of the highest reported rates of ESBL-Eb in the world. Actually, studies in CAR consistently show high prevalence of ESBL-Eb. A retrospective analysis of *Enterobacteriaceae* causing urinary tract infection isolated at the Institut Pasteur de Bangui showed an increase of the ESBL isolates from 3.7% to 19.3% between 2004 and 2006 [27]. A more recent survey in the same laboratory analyzed 941 *Enterobacteriaceae* isolated between May 2018 and April 2019 and found 478 (50.8%) ESBL clones [28]. Another study analyzing fecal carriage of healthy children below 5 years of age recruited between December 2011 and November 2013 reported 59% ESBLs among the *Enterobacteriaceae* [26]. Our results indicate an increasing trend of ESBL-Eb circulating in Bangui, CAR. There are several reasons that could explain this phenomenon, including auto-medication, acquisition of antibiotics without medical prescription, poor hospital hygiene that favors the selection of resistant bacteria, and empiric antibiotic treatment without antimicrobial susceptibility testing. These are common causes of AMR increase and dissemination in LMIC. Interestingly, a study analyzing AMR in *Enterobacteriaceae* isolated from fecal samples of wild animals living in the Dzanga-Sangha Protected Area, a national park and dense forest area with limited human access, reported 10% ESBL-Eb [29]. This result strongly supports human activities and practices as the main cause of the high rate of ESBL-Eb found in hospitals and communities in CAR. Strong actions are needed to address AMR increase in CAR, including capacity building for AST, policies regulating the use of antimicrobials, and sustained surveillance programs for early detection of potential outbreaks due to MDR clones. Another important finding of this study is the prevalence of clavulanic acid-resistant ESBL isolates (Table 6). Indeed, ESBL-Eb were originally defined as resistant to third- and fourth-generation cephalosporins and susceptible to BL inhibitors. With the emergence of inhibitor-resistant clones, this definition needs to be altered. Inhibitor-resistant ESBL-Eb represent a concern, since they contribute to limiting treatment options. The prevalence of these bacteria is unknown in most countries in sub-Saharan Africa. It is of importance to study the prevalence of these bacteria and identify the genes responsible for this resistance. A genomic surveillance based on whole genome sequence analysis constitutes a good approach for such an endeavor. Sub-Saharan African countries need to build capacity in this area to properly address the threat posed by ESBL bacteria and AMR in general.

## 4. Materials and Methods

### 4.1. Type and Population of Study, and Data Collection

We conducted a prospective cross-sectional cohort study to determine the fecal carriage of antibiotic-resistant *Enterobacteriaceae* in 0–5-year-old children admitted to the Pediatric University Hospital Complex, Bangui, CAR, during the period of July–September 2021. We included children independently of their disease condition. The study was approved by the ethical and scientific committee of the Faculty of Health Sciences, University of Bangui, and the Institut Pasteur de Bangui. Parents or guardians of children of both genders who had not received antibiotic treatment in the last 30 days were approached, and participants enrolled after obtaining informed consent.

### 4.2. Specimen Collection, Bacterial Isolation and Identification

Fecal samples were collected in dedicated boxes by the parents or guardians, placed in a cooler and transported to the bacteriology laboratory in less than two hours for immediate processing. Depending on the aspect of the stools, the samples were directly inoculated onto bromocresol purple (BCP) agar (Bio-Rad, Marnes la Coquette, France) and CHROMagar™ ESBL agar (Bio-Rad, Marnes la Coquette, France) plates or diluted in 1.5 mL saline before inoculation.

After overnight culture, plates were examined for bacterial growth. Depending on colony morphology, different morphotypes were re-isolated on the same plates and incubated for 16–24 h. Well-isolated bacterial colonies were characterized by Gram staining, oxidase and urease tests before identification using API 20E galleries, a standardized identification system for Enterobacteriaceae. API 20E results were interpreted with API Web Standalone 1.3.2. software (www.biomerieux.com). In the absence of growth in both BCP and CHROMagar plates, the samples were considered as negative. When growth occurred in only one of the two media (BCP or CHRO-Magar), the corresponding colonies were re-isolated on the same plates and further characterized by species identification and AST. In cases of growth in both BCP and CHROMagar, the corresponding colonies were first re-isolated on the same plates, then further characterized, and the AST profiles were compared. Colonies from the same sample growing in both media, belonging to the same species and giving a similar AST profile were considered as a unique clone and were stored and recorded as a single isolate to avoid analysis of duplicates.

### 4.3. Antimicrobial Susceptibility Testing

Antimicrobial susceptibility testing was performed using the Kirby Bauer disc diffusion method [30] on Mueller-Hinton agar plates. The antibiotic discs (Bio-Rad Antibiotic Disks, Marnes la Coquette, France) used included ampicillin (30 μg), ticarcillin (75 μg), cefalexin (30 μg), ceftriaxone (30 μg), cefotaxime (5 μg), cefepime (30 μg), ceftazidime (10 μg), amoxicillin/clavulanic acid combination (20–10 μg), cefoxitin (5 μg), tobramycin (10 μg), gentamicin (10 μg), netilmicin (10 μg), ciprofloxacin (5 μg), nalidixic acid (30 μg), chloramphenicol (30 μg), imipenem (10 μg), ertapenem (10 μg), amikacin (30 μg) and tigecycline (15 μg). The results were interpreted following the recommendations of the European Committee on Antimicrobial Susceptibility Testing (EUCAST) 2021 and were recorded as susceptible, intermediate or resistant. MDR strains were clones resistant to at least one molecule of at least three antibiotic classes. The ESBL phenotype was confirmed for isolates grown on CHROMagar ESBL or isolated on BCP by testing their resistance to third- and fourth-generation cephalosporins, including cefotaxime, ceftazidime, ceftriazone and cefepime. Additionally, the double-disc synergy test was performed using an amoxicillin/clavulanic acid combination. A positive synergy test was recorded as a confirmation of the ESBL-producing phenotype. An isolate grown on a CHROMagar plate that was resistant to third- and fourth-generation cephalosporin and to amoxicillin/clavulanic acid combination was considered as an inhibitor-resistant ESBL clone.

### 4.4. Statistical Analyses

Statistical analyses was performed using IBM SPSS v28 software. Pearson's correlation test was used to test for association of bacterial carriage with disease, age and gender. Pearson Chi-Square was used to compare antibiotic resistance between ESBL-Eb and non-ESBL-Eb, with a $p$ value < 0.05 being considered as statistically significant.

## 5. Conclusions

This study suggests a continuing rise of the prevalence of ESBL-Eb in Bangui, CAR. The fecal carriage of these bacteria, their similar distribution in the participants and the high proportion of 0–1-year-old children suggest that the rate of ESBL-Eb we found reflects the prevalence in the community. A strong surveillance system based on a one-health approach is needed to identify the source of contamination and the dissemination routes and prevent their spread in healthcare settings. In addition, it will be important to conduct genomic analyses of the ESBL-Eb circulating in Bangui in order to obtain insights into their pathogenic potential and prepare for future outbreaks.

**Supplementary Materials:** The following supporting information can be downloaded at: https://www.mdpi.com/article/10.3390/bacteria2010005/s1, Table S1: resistance profile of *Enterobacteriaceae* isolated from children at the Pediatric Hospital Complex, Bangui.

**Author Contributions:** Conceptualization, H.S.-W., J.-C.G. and E.K.K. Investigation, H.S.-W., J.-C.G., E.K.K., G.N. and J.-R.M. Formal analysis, H.S.-W., G.N., J.-R.M., B.M.Y., A.M. and Y.D. Validation, B.M.Y. and A.M. Supervision, C.F. and Y.D. Original draft, Y.D. Review and editing, H.S.-W., C.F. and A.D. Funding acquisition, G.V. Resources, G.V., A.D. and Y.D. All authors have read and agreed to the published version of the manuscript.

**Funding:** This work was supported by the Institut Pasteur of Bangui, the Department of Cooperation and Cultural Action of the French Embassy in Bangui, and the Department of International Network Affairs of the Institut Pasteur of Paris. The funders had no role in the design of the study, the collection, analysis, and interpretation of the data, or in the writing of the manuscript.

**Institutional Review Board Statement:** This study was approved by the ethical and scientific committee of the Faculty of Health Sciences, University of Bangui, and the Institut Pasteur de Bangui, under permit No. 17/UB/FACSS/CES/21. Written informed consent was obtained from the parents or guardians on behalf of the children included in this study.

**Informed Consent Statement:** Informed consent was obtained from all subjects involved in the study. Written informed consent was obtained from the patient(s) to publish this paper.

**Data Availability Statement:** All relevant data of this study are presented in this article including the Supplementary Table.

**Acknowledgments:** We thank all the patients who contributed to this study with their specimens. We are grateful to the staff of the Pediatric University Hospital Complex of Bangui and the biologists and technicians of the Laboratory of Medical and Experimental Bacteriology, Institut Pasteur of Bangui.

**Conflicts of Interest:** The authors declare no conflict of interest.

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
