# Peer review of "High Fecal Carriage of Extended-Spectrum β-Lactamase Producing Enterobacteriaceae by Children Admitted to the Pediatric University Hospital Complex in Bangui, Central African Republic"

_2674-1334, doi:10.3390/bacteria2010005_

Round 1
Reviewer 1 Report
in the introduction the authors can explain what esbl genes could be expected (tem, shv, ctx, ges etc)
in the result section the results could be improved when imformation is given about the other resistances ( tobr, gent, netil, amik, cipr, chloram, tigecycline) split by esbl positive and negative children. (i could not download the supporting information, file not found)
i could not find a report on previous results on esbl carriage in small children, therefore the conclusion “ a continuing rise” maybe not correct ( it could be that there other data on small children)
Author Response
- Add the esbl genes that could be expected.
A paragraph was added in the Introduction section after line 54 of the original manuscript to expand on β-lactams, β-lactamase and ESBL.
- In the result section the results could be improved when imformation is given about the other resistances ( tobr, gent, netil, amik, cipr, chloram, tigecycline) split by esbl positive and negative children
Analysis of antibiotic resistance according to ESBL phenotype is added in subsection 2.3 of the Results section with an additional table (Table 4 of the revised manuscript) showing the results.
- I could not find a report on previous results on esbl carriage in small children, therefore the conclusion “ a continuing rise” maybe not correct ( it could be that there other data on small children).
Lines 157-164 of the Discussion section mention previous studies in Central Africa Republic with the proportions of ESBL Enterobacteriaceae isolated and the corresponding references (refs 16-18 of the original manuscript). These studies show a continuing rise of the proportion of ESBL-producing Enterobacteriaceae in Central Africa republic.

Reviewer 2 Report
In line 26 (abstract), the setence must be reformulated to give better undestending
Author Response
- Reformulate the sentence line 26 (Abstract) to give a better understanding”
The sentence line 26 was reformulated as requested.

Reviewer 3 Report
Thank you for giving me the opportunity to revise this interesting work by Sanke-Waïgana et al, where the authors present their study on the fecal carriage of ESBL-producing Enterobacteriaceae by children admitted at a pediatric hospital in Bangui, Central African Republic.
Some comments should, however, be addressed before considering publication:
Title and Abstact:
-. Please use “Enterobacteriaceae” instead of “enterobacteriaceae”, as it is a bacterial family and, so, the first letter should be a capital letter.
Introduction section:
-. Please revise reference No.2: Lord O’Neill’s review is dated in December 2014, but the reference in this work indicates 2016.
-. P2 line55: please consider using “Enterobacteriaceae” with a capital letter. Here and in the rest of the manuscript.
-. P2 line55: consider adding “an” before the word “important”.
-. P2 lines 57, 58 and 63: please type names of bacteria in italics.
-. P2 line 58: it is not correct to present E. coli here. The first time that a bacteria name is presented, it should be typed complete Genus+species (Escherichia coli instead of E. coli). Only once the complete name is presented, then, it must be written with the first letter of the genus (E. coli) and in italics font. Please revise this and other bacteria names in the document.
-. P2 line 61: use “genes” instead of the singular, as there is more than one ESBL gene.
-. P2 lines 67-68: I encourage the authors to add a reference that supports that statement.
Results section:
-. P2 line 80-81: please, provide the p-values of the Pearson’s correlation coefficients.
-. P3 lines 90-91: when I sum up the isolates of Escherichia coli from Table 2 (147+19+7+2+1) the result is 176 isolates, but the text depicts n=175. Please, revise. Likewise, the total number of Klebsiella pneumoniae (44+19) sums up 63, whereas the authors inform n=64 in the text. Please, clarify.
-. P3 line 90: please, replace “K. pneumonia” with “K. pneumoniae”.
-. P3 lines 92-95: the authors use the term “infection” in this paragraph, and there is somehow leading to confusion. As infection is associated with clinical signs, it would seem that all positive isolates were collected from children with gastrointestinal infection, but it is not what the manuscript presents because only 10.1% of the children presented gastrointestinal clinical syndromes. I strongly suggest the authors to reconsider the use of the optimal term in this paragraph, replacing “infection” by “carriage” / “co-carriage” or “mono-carriage” / “bi-carriage” instead, when appropriate.
-. Table 2: use italics for naming bacteria.
-. P3 lines 106-107: the sentence “MDR strains that are clones resistant to at least one molecule of at least three antibiotic classes” should be reworded and placed in the Methods section, not in the Results section, as it is a definition for an outcome of the study.
-. Table 3: Use “Extended spectrum beta-lactamase” instead of ESBL in the table title. Use a first capital letter in “Enterobacteriaceae” in the table title. Also, “Hospital” and “Complex” to be in line with the way it has been presented in the rest of the manuscript.
-. Table 4: Use a first capital letter in “Enterobacteriaceae” in the table title.
Discussion section:
-. P4 lines 131, 140, 143: Use “Enterobacteriaceae” with the first capital letter.
-. P4 line 148: Replace “K. pneumonia” with “K. pneumoniae”.
-. P4 line 148: As commented before, it should be avoided the use of “co-infections” in this study. Reconsider replacing “co-infections” with “co-carriage” also here.
-. P4 line 151: reference 14 lacks of the title (Trends in Human Fecal Carriage of Extended-Spectrum β-Lactamases in the Community: Toward the Globalization of CTX-M) of the manuscript referenced.
-. P5 line 163, 168: Use “Enterobacteriaceae” with the first capital letter.
Materials and Methods section:
-. P6 line 205: use “performed” instead of “perform”.
-. P6 lines 213-214: please use the correct terms for SIR: susceptible (instead of “sensitive”), and intermediate (instead of “intermediary”).
-. P6 line 216: same comment (susceptibility instead of sensitivity). See: https://journals.asm.org/doi/10.1128/JCM.01702-19 by the American Society for Microbiology.
Supplementary Table 1:
Although the supplementary material is responsibility of the authors, I strongly suggest revising and addressing several points in it:
-. Column 2: Unify the way antibiotics are presented: all either beginning with a capital letter, or all in lower case.
-. Column 2: Avoid using short names as “nal”, “cipro” and “chlora”, and replace them with the correct name.
-. Column 3: There are four cells with number within parenthesis, but there is no explanation about the information they are intended to add. Please, either provide information in a legend, or delete them.
-. All acronyms in the table should be explained in a legend (i.e. Kp, Kor, etc).
-. The colors seem to indicate antibiotic classes, but this fact is not clarified in the table, thus I suggest to indicate this below the table, or in a new column, following the WHO ATC classification.
Author Response
Responses to Reviewer 3
- Use Enterobacteriaceae instead of Enterobacteria throughout the manuscript
The word Enterobacteriaceae was used wherever relevant throughout the manuscript.
- Reference 2 date in 2014 and not 2016
We thank the reviewer for porting out this: actually, O’Neill and team published their study as a report for WHO in 2014 and as a journal article in 2016. We mentioned in the text the year 2014 and use the journal article (easier to insert using a reference manager) as a reference.
- P2, line 55: add “an” before “important”
This change was made.
- P2, lines 57, 58 and 63: bacteria names in italics
Done.
- P2, line 58: write Escherichia coli since this is the first time in the document; + put all bacterial names in italics in the whole document
Done.
- P2, line 61: add “s” to gene
Done.
- P2, lines 67-68: add a reference to support the statement
A reference was added to support the statement.
- P2, lines 81-82: give the p values of the Pearson correlations
The Pearson correlation coefficients and p values were added.
- P3, lines 90-91: the total number of E. coli and Kp should be 176 and 63 respectively instead of 175 and 64
The number and percentage of E. coli and K. pneumoniae isolates were corrected.
- P3, lines 92-95: replace the term “infection” with “carriage”
We agree with the reviewer that “carriage” is more relevant than “infection” and changes the words as required.
- Table 2: bacterial names in italics
Done.
- P3, lines 106-107: put the sentence in the Methods section instead of the Results section
The sentenced was moved to the Methods section of the revised manuscript.
- Table 3: do not abbreviate ESBL in the title; use capital initial for Hospital and Complex; + Enterobacteriaceae
The requested changes were made.
- Table 4: Enterobacteriaceae
Done
- P4, lines 131, 140, 143: Enterobacteriaceae
Done
- P4, lines 148: pneumoniae
Done
- P4, lines 148: carriage instead of infection
Done
- P4, lines 151: title is missing in ref 14
The title of reference 14 was added.
- P5, lines 163 and 168: Enterobacteriaceae
Done
- P6, line 205: performed
Done
- P6, line 213-214, 216: susceptible instead of sensitive and intermediate instead of intermediary
Done
- Supplementary Table: column 2 harmonize the writing of antibiotic names (capital initial for all) + write the full name
Done.
- Column 4: clarify the four cells with parenthesis meaning intermediate
Done.
- Explain all the acronyms (Kp, Kor, etc.) in a footnote
Done.
- Explain the meaning of the colors (antibiotic classes) and follow the WHO ATC classification
Done.

Reviewer 4 Report
The manuscript presents “High fecal carriage of extended-spectrum β-lactamase producing Enterobacteriaceae by children admitted at the Paediatric University Hospital Complex in Bangui, Central African Republic” The author explained the prevalence of ESBLs in Enterobacteriaceae that will be beneficent for researcher, physician, and others to overcome this issue and adopt an alternative way to combat these challenges. I am highlighting a few mistakes as an author should writhe the name of all bacteria in italic. Escherichia coli O157:H7 is a serotype of the bacterial species Escherichia coli and is one of the Shiga-like toxins–producing types of E. coli. E. coli 0157 is the only strain that is responsible to cause diarrhea. E. coli is the normal flora of the gut and comes into the stool. The author should address that which strain of E. coli was identified in stool samples.
The title should be concise, accurate, and informative. Please revise this accordingly.
Introduction:
· The introduction is too general: Report the epidemiology of carbapenemases and ESBLs that focus on the study area. Please show the study gaps.
· Line 25: First, please write down the name of all bacteria in italic. Secondly, use the full name of the bacteria first i.e., Staphylococcus aureus (S. aureus), and then you can use the short name of the bacteria i.e., S. aureus. Enterobacteriaceae?
· Line 26-31: Not clear. The author should rewrite and explain it clearly.
· Line 57: Salmonella and Shigella?
· Line 143-146: A strong statement “Thirdly, Enterobacteriaceae are 143 powerful vectors of AMR spread since the gastrointestinal tract houses a luxurious flora 144 that horizontally exchange genetic elements including vectors of antimicrobial resistance 145 determinants” is given without reference. Please give an appropriate reference.
Results:
· Section 2.1: What is the correlation of anemia and respiratory illness with fecal ESBLs as mentioned in the title?
· Table 1: What is meant by “Others”? Please explain it in the legend of this table.
· The author should present the data of antimicrobial susceptibility testing as mentioned in the abstract that 19 antibiotics were tested against bacteria.
· The author should present the data in a simple way.
· The discussion is not clear. The author should make a fruitful discussion by using recently conducted studies. Due to this fact, I recommend the authors to read and add the following papers to the References section of the manuscript to have fruitful Introduction and Discussion sections: doi.org/10.1155/2022/5727638, doi.org/10.3390/pathogens11091019, doi.org/10.1128/spectrum.02137-22, 10.1007/978-3-030-76320-6_20, 10.3390/antibiotics12010029
Conclusion:
· Conclusion is not coming from results and Discussion. Please revise this.
Material and Methods:
· Section 4.1:Please give the description of the Hospital. What is the bed capacity? Please refer the SROBE checklist to ensure each section of the manuscript is well written (https://www.strobestatement.org/checklists/)
· Section 4.2: Not clear regarding sample collection, transportation, and preservation: How did you collect the blood sample? How did you transport the samples into the microbiology lab? Did you preserve the sample? If yes, then for how many days, and what was the temperature?
· Provide the manufacturer details of each chemical/material used (Company, City, Country).
· BCP is not defined prior to using the acronym.
· Please give a reference why the author only use BCP agar and CHROMagar™ ESBL agar for fecal sample inoculation.
· Line 194: API 20E is not defined prior to using the acronym.
· Section 4.3: The author should explain about disk diffusion method either used for measurement of a zone of inhibition or etc.
· The author should explain how the ESBLs were identified in this study.
Author Response
Responses to Reviewer 4
I am highlighting a few mistakes as an author should writhe the name of all bacteria in italic.
All bacteria names are now written in italics.
Escherichia coli O157:H7 is a serotype of the bacterial species Escherichia coli and is one of the Shiga-like toxins–producing types of E. coli. E. coli 0157 is the only strain that is responsible to cause diarrhea. E. coli is the normal flora of the gut and comes into the stool. The author should address that which strain of E. coli was identified in stool samples.
We are very well aware that most E. coli are commensals of human gut and that some strains are pathogenic, O157:H7 being one of them (but not the only one). Identifying the serotypes of the recovered bacteria was not part of the work we intended to do. We were interested in antibiotic resistance, especially ESBL production by the recovered Enterobacteriaceae. Therefore, the serotypes of the isolates were not investigated.
The title should be concise, accurate, and informative. Please revise this accordingly.
In our opinion, the title exactly reflect the scope of the work. We could not possibly suggest a better one. We would be thankful to the reviewer to suggest a better title.
The introduction is too general: Report the epidemiology of carbapenemases and ESBLs that focus on the study area. Please show the study gaps.
A paragraph was added in the Introduction section after line 54 of the original manuscript to expand on β-lactams, β-lactamase and ESBL.
Line 25: First, please write down the name of all bacteria in italic. Secondly, use the full name of the bacteria first i.e., Staphylococcus aureus (S. aureus), and then you can use the short name of the bacteria i.e., S. aureus.
We thank the reviewer for pointing out these errors that were corrected in the whole manuscript.
Line 26-31: Not clear. The author should rewrite and explain it clearly.
We thank the author for suggesting this clarification. This part of the abstract was modified.
Line 57: Salmonella and Shigella?
We are mentioning Salmonella and Shigella as examples of enteric bacterial pathogens.
Line 143-146: A strong statement “Thirdly, Enterobacteriaceae are powerful vectors of AMR spread since the gastrointestinal tract houses a luxurious flora that horizontally exchange genetic elements including vectors of antimicrobial resistance determinants” is given without reference. Please give an appropriate reference.
A reference supporting the mentioned statement is given.
Section 2.1: What is the correlation of anemia and respiratory illness with fecal ESBLs as mentioned in the title?
There is no correlation between these diseases and fecal carriage of ESBL-producing Enterobacteriaceae. As mentioned in the Discussion section (lines 133-136 of the original version) we did not intend to link presence of EBSL-Eb and disease condition. We added this in the Materials and methods section for clarification.
Table 1: What is meant by “Others”? Please explain it in the legend of this table.
We replaced the term “Others” with “Other diseases” in the corresponding case of Table 1.
The author should present the data of antimicrobial susceptibility testing as mentioned in the abstract that 19 antibiotics were tested against bacteria.
Data on resistance to the 19 tested antibiotics are shown in Supplementary Table 1.
The discussion is not clear. The author should make a fruitful discussion by using recently conducted studies. Due to this fact, I recommend the authors to read and add the following papers to the References section of the manuscript to have fruitful Introduction and Discussion.
We kindly disagree with the reviewer in that the discussion is not clear. This study was undertaken to investigate about the rise of ESBL-Eb in CAR observed from previous investigations. We wanted to have recent insights on this and indeed confirmed a high prevalence of ESBL-Eb in children that likely reflect what is happening in the communities. We discussed the possible causes of this phenomenon and suggest strategies to address it.
Conclusion is not coming from results and Discussion. Please revise this.
The conclusion gives three messages, (i) evidence of a continuing rise of ESBL-Eb in CAR, (ii) the need of a surveillance system and (iii) the suggestion to use genomic analysis to further characterize ESBL-Eb and prepare for outbreaks. All these come from the study including the discussion section.
Please give the description of the Hospital. What is the bed capacity? Please refer the SROBE checklist to ensure each section of the manuscript is well written (https://www.strobestatement.org/checklists/).
We thank the reviewer for suggesting to follow STROBE’s recommendations. However, this is not an observational study. As stated in the manuscript, our goal was to analyze fecal carriage of ESBL-Eb and we used a hospital setting as a means to access to children since our laboratory in Bangui and the children’s hospital have been collaborating for long. In this regard, details regarding the hospital are out of the scope of our study.
Section 4.2: Not clear regarding sample collection, transportation, and preservation: How did you collect the blood sample? How did you transport the samples into the microbiology lab? Did you preserve the sample? If yes, then for how many days, and what was the temperature?
Details about fecal sample collection and transportation were given lines 186-188 of the initial manuscript. No blood sample was collected only stools were obtained. Also, samples were transported to the laboratory within two hours after collection (lines 187) and immediately processed. All the samples were processed the day of their collection. Please note that the hospital and the Institut Pasteur de Bangui where the laboratory is located are contiguous.
Provide the manufacturer details of each chemical/material used (Company, City, Country).
Details on the products used were added.
BCP is not defined prior to using the acronym.
Done.
Please give a reference why the author only use BCP agar and CHROMagar™ ESBL agar for fecal sample inoculation.
CHROMagarTM-ESBL is a selective medium designed for isolation of ESBL-producing Gram-negative bacteria. BCP is a non-selective medium that support growth of Enterobacteriaceae and additionally detects the ability of these bacteria to metabolize lactose. By combining these two media we were able to efficiently isolate Enterobacteriaceae from stools and identify ESBL-producing clones. However, it should be pointed that other approach could be used. We used BCP and CHROMagar since they were satisfactory for what we wanted to do.
Line 194: API 20E is not defined prior to using the acronym.
We added a sentence to explain about API 20E galleries.
Section 4.3: The author should explain about disk diffusion method either used for measurement of a zone of inhibition or etc.
The Kirby Bauer disc diffusion method is a widely used and well-known technique. We added a reference about this technique.
The author should explain how the ESBLs were identified in this study.
We added additional details on the identification of ESBL-producing isolated in sub-section 4.3 of the Materials and methods section

Round 2
Reviewer 3 Report
Thank you for addressing the comments and suggestions. Now I find the article improved and I accept the revision.
Reviewer 4 Report
The author has addressed my all comments. The revised manuscript looks good.